Effects of clay materials and moisture levels on habitat preference and survivorship of Formosan subterranean termite, Coptotermes formosanus Shiraki (Blattodea: Rhinotermitidae)

Jin Zhengya 1
Chen Jian 2
Wen Xiujun 1
Wang Cai wangcai@scau.edu.cn 1
1 Guangdong Key Laboratory for Innovation Development and Utilization of Forest Plant Germplasm, College of Forestry and Landscape Architecture, South China Agricultural University , Guangzhou , Guangdong , China
2 Biological Control of Pests Research Unit, Agricultural Research Service, United States Department of Agriculture , Stoneville , MS , United States of America
Negri Ilaria
Electronic publication date: 2020 Oct 28
Publication date: 2020
Volume: 8
Electronic Location ID: e10243
Received 2020 Jun 26; Accepted 2020 Oct 5
Copyright: ©2020 Jin et al.
Copyright year: 2020
Copyright holder: Jin et al.
License: This is an open access article distributed under the terms of the Creative Commons Attribution License, which permits unrestricted use, distribution, reproduction and adaptation in any medium and for any purpose provided that it is properly attributed. For attribution, the original author(s), title, publication source (PeerJ) and either DOI or URL of the article must be cited.
License URL: https://creativecommons.org/licenses/by/4.0/

Keywords: Soil organism, Clay-insect interaction, Bentonite, Chlorite, Attapulgite

Funding: National Natural Science Foundation of China 31500530 This research was funded by the National Natural Science Foundation of China (31500530). The funders had no role in study design, data collection and analysis, decision to publish, or preparation of the manuscript.

==============================
Formosan subterranean termites, Coptotermes formosanus Shiraki, usually transport clay materials into tree hollows and bait stations. Our previous research showed that C. formosanus preferred to aggregate in the locations containing field-collected clay samples, but it was not clear whether this preference was influenced by clay types and/or moisture. In the present study, we conducted multiple-choice tests under low-moisture (25% moisture) or moderate-moisture (50% moisture) conditions to evaluate the aggregation and wood-feeding preferences of C. formosanus responding to hollow wooden cylinders (simulation of tree hollows) or baiting containers (simulation of bait stations) filled with different clay materials (bentonite , kaolin, chlorite, illite, or attapulgite), soil, or unfilled. Under low-moisture conditions, the majority of termites were found in the wooden cylinders or baiting containers filled with bentonite. Under moderate-moisture conditions, however, termites preferred to aggregate in wooden cylinders filled with chlorite or attapulgite; the percentages of termites that stayed in baiting containers filled with chlorite, attapulgite or soil were similar, which were significantly higher than those that filled with kaolin, illite, or unfilled. We then conducted no-choice tests to study the effect of clay materials on termites. Under low-moisture conditions, clay filled in the baiting containers significantly increased survivorship and body water percentage (an indicator of termite vigor) of termites, whereas no similar effect was detected under moderate-moisture conditions. This study demonstrated that both clay type and moisture affect termites’ preference.

Introduction

The Formosan subterranean termite, Coptotermes formosanus Shiraki (Blattodea: Rhinotermitidae), is an economically significant pest distributed in many warm temperate/subtropical regions of the world, including the United States of America, China, and Japan (Su, 2003; Austin et al., 2006; Scheffrahn et al., 2015; Chouvenc, Scheffrahn & Su, 2016). Suszkiw (2000) estimated that the annual repair and control cost of this pest was ∼1 billion dollars in the United States. C. formosanus damages not only wooden structures, but also live plants (Lai et al., 1983; Evans, Forschler & Trettin, 2019). Lai et al. (1983) reported that C. formosanus usually consumed the xylem, tree vascular tissue that transports water with dissolved minerals from the roots to stems and leaves and provides physical support as well. Although these trees may appear healthy and normal (the living part of the tree remained undamaged), they could be easily broken by winds (Lai et al., 1983). C. formosanus can also damage trunks and roots, which may result in wilt and sometimes death of the plant (Lai et al., 1983). The impacts of C. formosanus on forests had been largely ignored in the past but have received some attention in recent years. For example, Evans, Forschler & Trettin (2019) reported that 38% experimental patches of forests around Charleston (South Carolina, USA) and 42% patches around New Orleans (Louisiana, USA) were infested by C. formosanus, which caused significantly more and larger tree hollows compared with patches without termite infestation.

Interestingly, C. formosanus usually transport large amounts of clay and soil into tree holes and other hollow spaces within/around food (e.g., gaps between bait stations and matrices) during foraging (Fig. 1). Henderson (2008) reported that clay was more preferred than soil for C. formosanus to fill the void space of tree holes, and a large amount of clay (∼1 m3) can be found in such voids. Wang, Henderson & Gautam (2015) found that C. formosanus preferred to aggregate in chambers with field-collected clay. Also, termites preferentially used clay to cover the smooth surface of containers and construct shelter tubes (Wang & Henderson, 2014; Wang, Henderson & Gautam, 2015). Xiong et al. (2018a) reported that significantly more C. formosanus and Reticulitermes guangzhouensis Ping aggregated and fed in baiting containers filled with bentonite, a clay mineral, than unfilled containers. In each of these studies, preference of termites to a single source of clay material was investigated and confirmed, but two questions remained unanswered: (1) Do clay types and moisture conditions influence the aggregation and feeding preference by termites? (2) Do clay materials benefit C. formosanus by improving their survival, vigor, and wood feeding?

Figure 1 Coptotermes formosanus usually transports clay or soil into tree holes (A) and bait stations (B).

The photograph in A was taken by Wenquan Qin, and the photograph in B was taken by Zhengya Jin.

Based on the structure and mineralogy, clay can be divided into several mineral groups including the montmorillonite/smectite group, kaolin group, chlorite group, illite group, and palygorskite group. In this study, five common clay materials were selected from these groups to investigate their effects on the survival and behavior of C. formosanus (Table 1). Multiple-choice tests were conducted to study the aggregation and wood-feeding preference of C. formosanus among different clay materials that were either filled in the hollow wooden cylinders (simulation of tree hollows) or baiting containers (simulation of baiting stations). Also, no-choice tests were conducted to investigate whether each clay material affects the survival, wood-feeding behavior, and vigor (indicated by body water percentage) of termites. All choice and no-choice tests were conducted under low-moisture (25% moisture) conditions that were relatively dry for termites, or moderate-moisture (50% moisture) conditions that were suitable for termites. Here we did not conduct experiments under high-moisture conditions because termites cannot make tunnels in substrates with extremely high moisture content (Gautam & Henderson, 2011a).

Table 1 Basic information about soil and clay used in the present study.

Clay/ soil	Clay mineral group	PH	Organic matter content (g/kg)	Total nitrogen content (g/kg)	Exchangeable cation content (cmol/kg)	CECa (cmol/kg)	Water absorption at each water saturation	
					K+	Na+	Ca2+	Mg2+		100%b	50%b	25%b	
Bentonite	Montmorillonite/ Smectite Group	9.47	4.03	0.13	1.52	54.96	13.93	1.22	54.98	8.14c	4.07c	2.03c	
Kaolin	Kaolin Group	9.74	0.46	0.02	0.03	0.06	5.86	3.33	1.00	0.70	0.35	0.17	
Chlorite	Chlorite Group	7.82	4.23	0.16	0.09	0.09	15.86	7.24	1.03	0.92	0.46	0.23	
Illite	Illite Group	8.09	2.02	0.23	3.73	0.30	0.49	0.13	5.35	1.09	0.55	0.27	
Attapulgite	Palygorskite Group	10.12	3.58	0.26	0.89	27.43	16.24	5.01	38.81	2.92	1.46	0.73	
Soil	–	4.83	13.48	0.60	0.04	0.01	0.16	0.03	5.55	0.41	0.20	0.10	
Notes.

a CEC indicates cation exchange capacity.

b Water saturation level.

c Amount of deionized water (g) can be absorbed by 1 g dry powder of clay to reach the 100%, 50%, or 25% water saturation level.

Materials and Methods

Termite collection and maintenance

Underground baiting stations that included 6 pine-wood sticks (without termiticide; 4.5 × 3.0 × 16.0 cm) surrounded by plastic collars (height: 19.5 cm; diameter of upper side: 15.0 cm; diameter of bottom side: 13.5 cm) were buried in different locations of the arboretum of South China Agricultural University (SCAU, Guangzhou, China). C. formosanus infestation in each bait station was checked monthly, and the station with large numbers of termites were brought to the laboratory (Xiong et al., 2018a). Three colony groups of C. formosanus were collected in November 2018 (for experiments 1 and 2), and two colony groups were collected in May 2019 (for experiment 3). Colonies were over 1 km apart (Table 2). Collected baiting stations were maintained in the lab in 60 L plastic storage boxes for <1 month (23−27 °C under darkness). Before the experiment, wood sticks were taken out and gently knocked to extract termites, and the caste distribution of each colony group were determined by counting 100 termites for 5 times (Table 2).

Table 2 Basic information of the three colony groups of Coptotermes formosanus used in the choice and no-choice tests.

Colony group	Collection site	Caste distributiona (%)	
		Choice test	No-choice test	
		Worker	Soldier	Worker	Soldier	
1	23°09′37″N, 113°21′30″E	99	1			
2	23°09′33″N, 113°21′07″E	97	3	99	1	
3	23°09′34″N, 113°21′08″E	94	6	96	4	
Notes.

a The percentages of workers and soldiers in each colony group were determined by counting 100 termites for five times. The mean percentages are shown here and used to set the experiment.

Soil and clay

Top soil (depth < 5 cm) was collected from a location (23°09′26″N, 113°21′08″E) in the arboretum of SCAU where C. formosanus activities have been detected. A sample of soil was sent to the Laboratory of Forestry and Soil Ecology (College of Forestry and Landscape Architecture, SCAU), and identified as sandy loam soil (12% clay, 21% silt, and 67% sand). The dried soil was ground with a wooden mortar and pestle, and then sifted through a 2-mm sieve to remove coarse particles and plant roots. Five clay materials, bentonite (Fresh & Natural™, Bentonite Performance Minerals LLC, Houston, USA), kaolin (Jufeng®, Shanxi Jufeng Kaolin Co., Ltd., Jinzhong, China), chlorite (Hetalc®, Haicheng Hetalc Powder Technology Co., Ltd., Haicheng, China), illite (Junhong®, Huiyuan Junhong New Material Co., Ltd, Datong, China), and attapulgite (Dingbang®, Dingbang Mineral Products Technology Co., Ltd., Changzhou, China), were purchased. Sample of clay materials or soil were sent to the Laboratory of Environmental Chemistry (College of Natural Resources and Environment, SCAU) to measure chemical properties (i.e., pH, contents of organic matter, total nitrogen, and exchangeable cation, and cation exchange capacity, as shown in Table 1). Clay and soil were sterilized at 80 °C for 3 days, and then completely dried at 50 °C for >2 weeks. The formula provided by Chen & Shelton (2017) was used to determine the moisture content of soil and clay: moisture content (%) = [weight of distilled water added/(weight of saturated soil or clay − weight of dried soil or clay)] × 100%. To prepare soil and clay with 25%- or 50%-moisture, the required amount of deionized water and dried soil or clay were placed in the zip-lock bags and thoroughly mixed (Table 1).

Experiment 1: preference of clay materials filled in wooden cylinders

Multiple-choice tests were conducted to study whether C. formosanus prefer to aggregate and feed in hollow wooden cylinders which were filled with certain clay material or soil, or unfilled. Each test (under either low- or moderate-moisture conditions) was repeated 12 times (each of the three colony groups was repeated 4 times). The bioassay arenas were plastic containers (volume = 1,250 mL, height = 6 cm, diameter of upper side = 19.2 cm, diameter of bottom side = 16 cm). To set the bioassays under low-moisture conditions, 25%-moisture soil (substrate) was added into the container to the depth of 0.5 cm. Hollow wooden cylinders (Schima superba Gardn. et Champ., height = 28 mm, internal diameter = 26 mm, external diameter = 30 mm, thickness of the bottom side = 3 mm, diameter of hole on the bottom side = 16 mm; Fig. 2A) were oven-dried at 80 °C for 5 days and weighed. Seven wooden cylinders were filled with 25%-moisture clay materials (bentonite, kaolin, chlorite, illite, or attapulgite) or soil, or unfilled (Fig. 2A). These wooden cylinders were placed on the substrate in the arena with randomly assigned orders, and the adjacent cylinders were equally distanced. Substrate (25%-moisture soil) was then added in the container until the base of cylinders was partially buried in the depth of 1 cm (Fig. 2B). Similar procedures were carried out to set the bioassays under moderate-moisture conditions, but clay and soil with 50% moisture were used.

Figure 2 Procedures to set experiment 1.

Hollow space of wooden cylinders was either filled with the clay material (chlorite, attapulgite, bentonite, kaolin, or illite) or soil, or remained unfilled (A). Substrate (soil) was added into the container to the depth of 0.5 cm, and seven wooden cylinders were placed on the substrate with randomly assigned orders. Additional substrate was then added until the base of cylinders was partially buried in the depth of 1 cm (B). The photographs were taken by Zhengya Jin.

Four hundred termites (percentages of workers and soldiers were determined by the caste distribution of each colony group as shown in Table 2) were counted and released into the center of each arena. The containers were sealed with plastic wrap to keep the internal moisture conditions and maintained in a 25 °C environmental chamber under total darkness. At the end of the experiment (day 21), we counted the number of live termites stayed within each wooden cylinder or substrate (none of the termites stayed on the outer wall of the wooden cylinders or on the surface of the substrate). The wooden cylinders were carefully washed using distilled water to remove any particles, and placed in an oven (80 °C) for 5 days and weighed. The dry weight change of wooden cylinders before and after the experiment was measured to calculate wood consumption.

Experiment 2: preference of clay materials filled in baiting containers

The protocols of multiple-choice tests provided by Xie et al. (2019a) were modified to investigate whether C. formosanus prefer to aggregate and feed in the baiting containers filled with each clay material or soil, or unfilled. Each choice-test (under either low- or moderate-moisture conditions) was repeated 12 times (each of the three colony groups was repeated 4 times). The bioassay arena was a 1250 mL plastic container as mentioned earlier. The baiting container was plastic box (height = 33 mm; diameter of upper side = 41 mm, diameter of bottom side = 31 mm) with 10 holes (5 mm in diameter, staggered distributed in two rows) drilled on the wall (Fig. 3A). A pine wood block (20 × 20 × 20 mm) was completely dried in an 80 °C oven for 5 days and weighed, and then placed into the box (Fig. 3A). Under low-moisture conditions, the void space of seven baiting containers was either filled with 25%-moisture clay (bentonite, kaolin, chlorite, illite, or attapulgite) or soil (Fig. 3B), or unfilled. These baiting containers were placed in the bioassay arenas with randomly assigned orders. Each baiting container was equally distanced with the adjacent ones. Substrate (25%-moisture soil) was then added until the baiting containers were buried (Fig. 3C). We carried out similar procedures to set the choice test under moderate-moisture conditions, but clay and soil at the 50%-moisture level were used.

Figure 3 Procedures to set experiment 2.

The baiting containers were prepared by drilling 10 holes on the wall of a plastic box and placing a wood block inside. Baiting containers was either remained unfilled (A), or filled with the clay material (bentonite, kaolin, chlorite, illite, or attapulgite) or soil (B). These baiting containers were then placed in the bioassay arenas with randomly assigned orders, and substrate (soil) was added until the baiting containers were buried (C). The photographs were taken by Zhengya Jin.

Four hundred termites were released into the center of each arena. The bioassay arenas were sealed with plastic wraps and placed in the environmental chamber (25 °C and under total darkness). At the end of the experiment (day 21), the number of live termites stayed within each baiting container or substrate was counted. The wood consumption was measured as mentioned earlier. The formula provided by Xie et al. (2019a) was used to calculate the moisture content of wood blocks: wood moisture (%) = [(wet weight of wood block after the experiment − dry weight of wood block after the experiment)/dry weight of wood block after the experiment] × 100%.

Experiment 3: effect of clay materials on survival, vigor, and wood consumption of termites

The protocols of no-choice tests provided by Xie et al. (2019a) were modified to investigate the survival, body water percentage (an indicator of termite vigor), and wood consumption of termites, as well as wood moisture when the baiting containers (as mentioned in experiment 2) were filled with each clay material (bentonite, kaolin, chlorite, illite, or attapulgite) or soil, or unfilled. The tests were conducted under either low-moisture (25% moisture) or moderate-moisture (50% moisture) conditions. In total, the tests contained 14 treatments, and each treatment was repeated 10 times (each of the two colony groups was repeated 5 times).

To set the bioassays under low-moisture conditions, a baiting container (either filled with 25%-moisture bentonite, kaolin, chlorite, illite, attapulgite, soil or unfilled) was placed in the center of a plastic box (volume = 300 mL, height = 62 mm; diameter of upper side = 86 mm, diameter of bottom side = 66 mm). Substrate (25%-moisture soil) was then added into the plastic box until the bating container was buried. The same procedure was conducted to set the bioassays under moderate-moisture conditions, but 50%-moisture soil or clay materials were used.

One hundred termites were counted and released into the center of each box. The bioassays were sealed with plastic wraps and placed in a completely dark environment chamber at 25 °C. The survivorship of termites was recorded at the end of the experiment (day 21). Body water percentage of termites were measured as previously described in Xie et al. (2019a). Specifically, 10 workers were randomly selected, and their fresh weight was measured using a 0.1 mg electronic balance. These termites were then placed in an oven (50 °C) for 3 days, and their dry weights were measured using the same electronic balance. The formula provided by Xie et al. (2019a) was used to calculate the body water percentage of termites: body water percentage (%) = [(fresh weight of termites − dry weight of termites)/fresh weight of termites] × 100%. The wood consumption and wood moisture were also measured and calculated as mentioned above.

Data analysis

For each multiple-choice test in the experiments 1 and 2, the percentage of termites that aggregated in each wooden cylinders or baiting containers (filled with bentonite, kaolin, chlorite, illite, attapulgite, soil, or unfilled), or stayed within the substrate was calculated. Because of the sum constraint of the percentage data, we applied log-ratio transformation mentioned by Kucera & Malmgren (1998) to make the raw percentages data independent. Two-way analysis of variance (ANOVA, SAS 9.4, SAS Institute, Cary, NC) was used to analyze the transformed data with termite colony as the random factor and aggregation site as the fixed factor. In addition, we compared wood consumption and wood moisture using the two-way ANOVA with termite colony as a random factor and filling types (filled with different clay materials or soil or remained unfilled) as a fixed factor. For no-choice tests in the experiments 3, we compared the survivorship, body water percentage, and wood consumption of termites, as well as wood moisture, using a two-way ANOVA with termite colony as the random factor and treatment as the fixed factor. Tukey’s Honest Significant Difference (HSD) test was used after each ANOVA for post-hoc comparisons. In all tests, the significance levels were determined at α = 0.05.

Results

Experiment 1: preference of clay materials filled in wooden cylinders

Under low-moisture conditions, low survivorship (<30%) of termites were found in the colony group 1. As a result, only data obtained from the colony groups 2 and 3 (survivorship ≥ 70%) were analyzed. Similar percentages of termites aggregated in the substrate and wooden cylinders filled with bentonite, both were significantly higher than that in other locations (Fig. 4A). No significant difference in wood consumption was detected when compared among wooden cylinders filled with each clay material or soil, or unfilled (Fig. 4B).

Figure 4 Results (mean ± SE) of experiment 1 (under low-moisture conditions) showing the percentage of termites in each location (A) and consumption of wooden cylinders (B).

Significant differences are indicated by different letters (P < 0.05).

Under moderate-moisture conditions, data obtained from the three termite colonies were analyzed. Similar percentages of termites were found in the wooden cylinders filled with attapulgite and chlorite, and both were significantly higher than that of wooden cylinders filled with kaolin, illite, or soil, or unfilled (Fig. 5A). However, wood consumption was similar when compared among filling types (Fig. 5B).

Figure 5 Results (mean ± SE) of experiment 1 (under moderate-moisture conditions) showing the percentage of termites in each location (A) and consumption of wooden cylinders (B).

Significant differences are indicated by different letters (P < 0.05).

Experiment 2: preference of clay materials filled in baiting containers

Under low-moisture conditions, low survivorship (<30%) of termites were found in the colony group 1, and therefore only data obtained from colony groups 2 and 3 (survivorship ≥ 70%) were analyzed. Most termites were found in the baiting containers filled with bentonite, whereas only a few termites were found in other locations (Fig. 6A). The wood consumption in the baiting containers filled with bentonite was also significantly higher than the other ones (Fig. 6B). The wood moisture was similar when compared among different filling types (Fig. 6C).

Figure 6 Results (mean ± SE) of experiment 2 (under low-moisture conditions) showing the percentage of termites in each location (A), and wood consumption (B) and wood moisture (C) in each baiting container.

Significant differences are indicated by different letters (P < 0.05).

Under moderate-moisture conditions, data obtained from the three termite colonies were analyzed. Similar percentages of termites were found in the baiting containers filled with attapulgite, chlorite, or soil, which were significantly higher than that of containers filled with kaolin or illite, or unfilled (Fig. 7A). The wood consumption was not significantly different among filling types (Fig. 7B), but wood moisture in baiting containers filled with bentonite was significantly higher than other ones (Fig. 7C).

Figure 7 Results (mean ± SE) of experiment 2 (under moderate-moisture conditions) showing the percentage of termites in each location (A), and wood consumption (B) and wood moisture (C) in each baiting container.

Significant differences are indicated by different letters (P < 0.05).

Experiment 3: effect of clay materials on survival, vigor, and wood consumption of termites

Under low-moisture conditions, treatments with clay materials had significantly higher termite survivorship compared to the treatment with unfilled baiting containers (Fig. 8A). Also, clay significantly increased body water percentage of termites compared to the treatment with soil-filled baiting containers under low-moisture conditions (Fig. 8B). No significant difference in survivorship and body water percentage of termites were detected among treatments under moderate-moisture conditions (Figs. 8A and 8B). Termites consumed least wood when baiting containers were filled with kaolin under low-moisture conditions, whereas the highest wood consumption was found when baiting containers were unfilled under moderate-moisture conditions (Fig. 8C). The wood moisture was significantly higher when the baiting containers were filled with bentonite, attapulgite, or chlorite (under moderate-moisture conditions) compared with the remained treatments (Fig. 8D).

Figure 8 Results (mean ± SE) of experiment 3 showing the survivorship (A), body water percentage (B), and wood consumption (C) of termites, and wood moisture (D) in each treatment.

Significant differences are indicated by different letters (P < 0.05).

Discussion

Many previous studies focused on clay preference and utilization by the higher fungus-growing termites in the family Termitidae. The biological functions of these termites as soil engineers have been reviewed by Bignell (2006), Pardeshi & Prusty (2010), Jouquet et al. (2011) and Jouquet et al. (2016). Jouquet et al. (2011) pointed out that higher termites transport large amounts of clay from various depths in underground sites to the soil surface. After they use clay to construct the aboveground biostructures such as sheetings and mounds, the subsequent erosion processes would modify the physical and chemical properties of surface soil, which may exert a significant impact on the ecosystems (Jouquet et al., 2011; Jouquet et al., 2016). For example, Harit et al. (2015) reported that the fungus-growing termite Hypotermes obscuriceps (Wasmann) constructed extensive sheetings on their food (leaves and branches), and when sheetings were degraded by rain a lot of inorganic ions (e.g., K+, F−, and Cl−) were released and therefore impacted soil properties. Sileshi et al. (2010) reported that mounds constructed by termites (e.g., species in the genus Ancistrotermes, Macrotermes, Odontotermes, Cubitermes, and Trinervitermes) were enriched in clay and nutrition, thus creating “islands of fertility” and enhancing the growth of plants, which shaped the spotted vegetation patterns in savannas across Africa. Bonachela et al. (2015) reported that such spotted vegetation patterns created by termites made the landscapes more robust to aridity and more stabilized to global climate changes. Likewise, Evans et al. (2011) reported that in the arid climate zones where earthworms are absent, termites provide alternative ecological services to improve water infiltration and soil nitrogen, and therefore contributed to the sustainability of dryland agriculture.

Compared to higher termites, much less attention has been paid to the potential interactions among clay, lower subterranean termites, and ecosystems. Harit et al. (2017) reviewed 29 articles about soil sheeting (always enriched in clay) produced by termites, and only 2 of them focused on the sheeting behaviors of lower subterranean termites, Psammotermes allocerus Silvestri (Vlieghe et al., 2015) and Coptotermes acinaciformis (Froggatt) (Oberst, Lai & Evans, 2016; Oberst et al., 2019). Vlieghe et al. (2015) observed a high level of soil or sand sheeting on grasses during the early formation stages of Namibian fairy circles constructed by P. allocerus. In contrast, the lowest number of termites and sheet grasses can be detected in the mature circles, which indicates that termites abandon their ephemeral polycalic nests within the mature circle (Vlieghe et al., 2015). Oberst et al. (2019) reported that C. acinaciformis performed clay-sheeting behaviors for different purposes according to the situational context. When the wood was unloaded, C. acinaciformis wrapped the dry wood with a layer of clay to increase its moisture levels. However, when the wood was loaded, C. acinaciformis kept the wood dry to improve its compressive strength and rigidity; meanwhile, they substituted some of the wood for clay walls to improve the bearing capacity. Some previous studies showed that C. formosanus also transported clay to fill the void volumes within tree holes or bait stations (Henderson, 2008). Such behaviors may result from tunnel excavation and subsequent transport processes (Li & Su, 2008; Lee et al., 2020). Some evidence also showed that these behaviors may be important for the foraging of lower subterranean termites because they preferred to aggregate and feed on food which was artificially covered/filled with clay (Wang & Henderson, 2014; Wang, Henderson & Gautam, 2015; Xiong et al., 2018a).

Previous studies showed that moisture conditions of substrates are important for the survival of subterranean termites (Sponsler & Appel, 1990; Cornelius & Osbrink, 2010; Gautam & Henderson, 2015). For example, Cornelius & Osbrink (2010) reported that C. formosanus contacted with dry soil had high mortality caused by desiccation. Gautam & Henderson (2015) reported that C. formosanus individuals showed three stages of desiccation—curling of antennae (stage I), on back but can right themselves and walk (stage II), and unable to get off back (stage III)—soon after exposed to open-air conditions, and most termites were destined to die once they reached stage III. Also, Gautam & Henderson (2015) reported a colony-level variation in the rate of water loss, since termites from one colony dried out faster than that from other colonies. Similarly, one of the three termite colonies used in our choice tests had a high mortality rate under low-moisture conditions, probably due to the low drought-tolerance of this colony. Interestingly, Gautam & Henderson (2015) showed that termites in the groups (n = 50) had a slower desiccation rate than the individual ones, and suggested that the large densities of field colonies of C. formosanus may contribute to maintaining high humidity levels and reducing water loss. One potential limitation of our studies is that we used only 400 termites in each replicate of the choice tests, and this relatively low density of termites may have an effect on the moisture-maintaining capacity and behaviors of termites, especially under low-moisture conditions. Future experiments with larger groups of termites would be helpful.

Our no-choice tests showed that all types of clay materials significantly increased body water percentage, an important indicator of termite vigor (Arquette et al., 2006), compared with soil under low-moisture conditions (Fig. 8B). This probably occurred because clay can retain more water than soil (as shown in Table 1), and created a more favorable microenvironment in the baiting containers. To avoid desiccation, C. formosanus has a strong tendency to aggregate in locations with proper humidity/moisture levels (Nakayama, Yoshimura & Imamura, 2005; Gautam & Henderson, 2011a; Gautam & Henderson, 2011b; Gautam & Henderson, 2011c). However, they do not prefer substrate and wood with extremely high moisture content (Nakayama, Yoshimura & Imamura, 2005; Gautam & Henderson, 2011a). Our choice tests showed that the majority of termites stayed in the wooden cylinders or baiting containers filled with bentonite under low-moisture conditions (Figs. 4 and 6). Such preference was moisture-dependent, because no significant termite aggregation was detected responding to the 50%-moisture bentonite. It is probable that termites can acquire enough water under moderate-moisture conditions. Therefore, it is not necessary to stay within bentonite to maintain moisture. Also, bentonite can absorb a large amount of water at the 50%-saturation level (Table 1), and caused much higher wood moisture compared with other clay materials or soil (Figs. 7C and 8D), which may inhibit the feeding activities of C. formosanus. Likewise, Carey et al. (2019) reported that the mound-building termites, Macrotermes michaelseni (Sjöstedt), performed different clay-relocation behaviors depending on the humidity levels, as they transport less soil and creates structures with smaller volumetric envelopes in the laboratory with low ambient humidity compared with high-humidity conditions.

The present study focused on the clay preference by termites under different moisture conditions. There may be other factors affecting termite behaviors in response to clay materials. For example, previous research showed that the lower subterranean termites Reticulitermes flavipes (Kollar) can detect certain ions in soil (Botch & Judd, 2011), and directly acquire micronutrients such as calcium, magnesium, iron, and manganese from the soil (Janzow & Judd, 2015). Our choice tests showed that termites tended to aggregate in wooden cylinders and baiting containers filled with attapulgite and chlorite under moderate-moisture conditions. These clay materials have high contents of exchangeable cations of calcium and magnesium (Table 1), which may result in the termite aggregation. In addition, previous studies showed that soil microbes such as Metarrhizium anisopliae (Metschn.) Sorok and Trichoderma fungi significantly influenced the aggregation and tunneling preferences of subterranean termites (Xiong et al., 2018b; Xiong et al., 2019; Wen et al., 2020). In our study, different microbes may colonize wooden cylinders or baiting containers containing each clay material and soil, and affect termites’ choice. Also, clay may have more biological functions for termites besides creating a moist environment and preventing termites from desiccation. For example, many social insects cover/fill food sources with various materials (e.g., soil particles) to block competitors and predators (e.g., Maciel et al., 2015; Mendonça, Santos-Prezoto & Prezoto, 2019; Qin et al., 2019). Chouvenc, Mullins & Su (2015) reported that when C. formosanus encountered its predator, the big-headed ant, Pheidole megacephala (Fabr.), termites immediately deposited particles and sealed the access point of ants to “create a physical separation with little to no casualties”. Clay materials in the void spaces may also protect termites from being exposed to the open-air environment and attacked by ants and other predators. One limitation of our study is that we only tested five clay materials, and many other clay or silt minerals may also affect aggregation and foraging behaviors of C. formosanus. It would be valuable to conduct choice and no-choice tests with more clay and silt materials in the future.

Baiting is one of the main methods to suppress and eliminate populations of subterranean termites (Evans & Iqbal, 2015; Su, 2019), but termites showed a low tendency to attack bait with low moisture levels (Cornelius & Osbrink, 2011). Our previous studies tried to create a microhabitat more suitable for termites by adding super absorbent polymers into bait stations. Although these water-retaining materials increased bait consumption, they did not cause significant aggregation by termites, probably because super absorbent polymers are gel-like and therefore termites cannot make tunnels to access the bait matrices (Xie et al., 2019a; Xie et al., 2019b). The present study showed that bentonite filled in the baiting containers not only caused aggregation of termites but also increased wood consumption under low-moisture conditions (Fig. 6A). This result indicated that placing bentonite in bait stations may increase termite infestation and bait consumption in drought locations. However, it is worth noting that moisture levels in reality may vary locally and fluctuate over time. Also, multiple environmental factors may influence termites’ choice in the field. It is important to conduct field studies to confirm whether bentonite can potentially be used as termite attractants under natural conditions.

Conclusion

In this study, we discovered that termites preferred to aggregate in wooden cylinders or baiting containers filled with bentonite compare with other clay materials (kaolin, chlorite, illite, and attapulgite) under low-moisture conditions. Also, termite consumed significantly more wood in the baiting containers filled with bentonite. Under moderate-moisture conditions, however, chlorite and attapulgite were most preferred by termites. In addition, no-choice tests showed that baiting containers filled with clay materials increased termite survivorship and body water percentage (an indicator of termite vigor) under low-moisture conditions. This study showed that both clay types and moisture are important for the survival and foraging activities of termites, and can influence their aggregation and wood-feeding preferences.

Supplemental Information

Data S1 Experiment 1 raw data, including the number of surviving termites, and the dry weight of wooden cylinders before and after the experiment

Click here for additional data file.

Data S2 Experiment 2 raw data, including the number of surviving termites, the dry weight of wooden blocks before and after the experiment, and the wet weight of wooden blocks after the experiment

Click here for additional data file.

Data S3 Experiment 3 raw data, including the number of surviving termites, the dry weight of wooden blocks before and after the experiment, and the wet weight of wooden blocks after the experiment, the fresh and dry weight of termites after the experiment

Click here for additional data file.

We sincerely thank Dr. Eric Riddick, Dr. Ilaria Negri (the editor), and three reviewers for their valuable comments and suggestions on the manuscript. We also thank Wenquan Qin (College of Forestry and Landscape Architecture, SCAU, Guangzhou, China) for providing the picture (Fig. 1A) used in this manuscript.

Additional Information and Declarations

Competing Interests

Author Contributions

Data Availability

The authors declare there are no competing interests.

Zhengya Jin performed the experiments, analyzed the data, prepared figures and/or tables, and approved the final draft.

Jian Chen analyzed the data, authored or reviewed drafts of the paper, and approved the final draft.

Xiujun Wen conceived and designed the experiments, authored or reviewed drafts of the paper, and approved the final draft.

Cai Wang conceived and designed the experiments, analyzed the data, prepared figures and/or tables, authored or reviewed drafts of the paper, and approved the final draft.

The following information was supplied regarding data availability:

Raw data for Experiments 1, 2, and 3 are available in the Supplemental Files.

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
