# Peer review of "Effects of clay materials and moisture levels on habitat preference and survivorship of Formosan subterranean termite, Coptotermes formosanus Shiraki (Blattodea: Rhinotermitidae)"

_PeerJ, doi:10.7717/peerj.10243_

## Round 0.1 · original submission · Minor Revisions

The manuscript is clearly written and, in my opinion, authors may address the reviewers’ requests without the need for running other experiments.

First of all, since no field-trials have been carried out, more caution should be put in drawing conclusions.

Authors should also better discuss the context of their experiments and provide a comprehensive discussion of other factors that have potentially influenced termites’ choice.

It should be also important to specify the physico-chemical properties and size of soil/clay used in the study (I think that specifics may be easily available from manufacturers), and why authors decided to choose such materials (or decided not to consider others, e.g. silt materials).
Please, also consider other relevant literature that may improve the discussion.

Reviewer 1 ·

Basic reporting

All of the main points are met. No comments.

Experimental design

No comments. All the points are sufficiently met.

Validity of the findings

No comments.

Additional comments

I found the manuscript to be clearly written, and the science overall relatively sound. I really could not find any critical issues. The one item that could be detailed better in the writing relates to use of the terms "dry" and "wet" to describe the moisture levels compared in experiments. 25% moisture is not necessarily "low" moisture, but is more intermediate, and in some ecosystem or soil types it can be the highest level encountered (e.g., sand). Better ways to refer to "dry" and "wet" throughout the manuscript would be low and high since dry and wet is less descriptive. In general I felt the experimental designs and setups were sound, and the interpretations relatively straightforward. The conclusions are appropriate based on the results. Previous reports are both acknowledged and appropriately discussed relative to the current results. I feel some minor revisions around how moisture levels are described should be sufficient to get the final paper ready for publication.

·

Basic reporting

This is fine. The English is good and there are no serious flaws with the structure or readability of the paper. I have identified below where some change of emphasis in the methods would be helpful

Experimental design

This is clear but complicated. I hope that the illustrations can go close to the methods, as a diagram is always useful for clarity.

Validity of the findings

The findings are interesting and add something useful to our knowledge of how termites move soil around. It is important, however, that a reader does not take away the impression that all lower termites move soil around. Only those that forage outside the nest do it. This means that the families Kalotermitidae, Archotermopsidae, and Stolotermitidae do not produce soil sheeting or shelter tubes.

Additional comments

A little more could be made about the choice of the minerals. At the moment, the introduction does not tell us anything about the clay minerals or their water-holding capacities. A few lines explaining why each clay mineral was chosen would help make the paper clearer

Reviewer 3 ·

Basic reporting

The use of English language is appropriate. Still, the manuscript could benefit from some more proofreading and logical consistency Eg.
- check for missed articles "the Unites States [of America], China,...", with "a wooden mortar and pestle"

- use "tree vascular tissue" instead of "plant"

- "was ground"

- "(none of the termites stayed on the outer wall of the wooden cylinders or on the surface of the substrate)"

- I think you should consider citing "Oberst et al, Termites manipulate moisture content of wood to maximise feeding, Biology Letters, 15(7). 20190365. https://doi.org/10.1098/rsbl.2019.0365"; since they considered similarly to you the influence of moisture levels on foraging of lower termites (Coptotermes acinaciformis). Also, when it comes to the discussion, this paper should be discussed and the lower and upper bounds of moisture levels relevant to termites (Gautam and Henderson need to be also more discussed in this context). T. A. Evans, T. Z. Dawes, P. R. Ward, N. Lo, Ants and termites increase crop yield in a dry climate, Nature Communications 2 (2011) 1–7 provide a valuable reference for ants and termites increasing soil quality especially in more arid climate zones.

Experimental design

I think the experiment is interesting since it focuses on clay properties, but some critical considerations were not taken, and the data /information provided is not fully discussed. Central in the experiment is the choice of clay materials and the number of termites chosen as well as the consequences resulting from that. Especially the novelty of the paper – which lies in using different clay materials is insufficiently presented and the analysis is not going deep enough. The number of termites and their behaviour and ecological and behavioural consequences due to the artificial situation constructed is not discussed in the extend required.

- Table 1 should be extended in that it is less of interest which company produces those clays, but rather what the different properties are, eg particle size, mineralisation, chemical composition and ionisation. Water retention has been reported, but then the table as well as all figures should be sorted according to the property investigated based on the hypothesis of the authors (eg water absorption)


- Also, the number of termites seem very small in order to generate a more natural condition – certainly 400 termites are very stressed and limiting the moisture results in changed behaviour. Also, the number of termites has an effect on the moisture in turn and this changes the behaviour – hence, separating single factors is difficult if not impossible. The authors should have shown the same effect for a larger number of termites (why is colony 1 not considered in experiment A? A clear sign of stress in termites).

- Please describe more in detail how you ensured the quality of the termites collected (I assume due to baiting only mature workers and soldiers were taken) and discuss why colony 1 in experiment A was not considered. I find the ratio very low 1 to 4 % soldiers only? This indicates to me that the baiting station was not property contacted yet (in lower termites of Cop. formosanus it is usually 5-10 % - and rather on the higher end towards 10% see eg The proportion of soldiers in termite colonies: a list and a bibliography (Isoptera) [1977]. Haverty, M.I

Validity of the findings

I do not doubt that the results are valid, but the findings need to be very critically discussed considering the situation / context of the experiments. More validation is required with regards to the clays used and more hypothesis should be developed based on other properties than moisture only. The authors should consider running a small field trial and monitoring their results and conditions using data loggers very carefully. Also an experiment with more termites and a larger arena could be used as alternative as I believe that in practice too many factors might influence the results so that the choice of termites in terms of clays cannot be observed. However, if in the field this preference can be observed it is a very strong indication for the correctness of the authors’ hypothesis.

- The authors discuss attapulgite and chlorite but why not bentonite? Certainly, the choice of termites is depending on the humidity levels and certain configurations have been found to be advantageous for termites (it is not ‘the more moisture, the better’ – see Gautam and Henderson (2010) or also Oberst et al (2019)).
- The conclusions drawn that bentonite can be used when its drier and attapulgite and chlorite if moist, is flawed, since moisture and humidity levels in reality fluctuate a fair bit and vary locally as well. I question therefore the practicality of the author’s hypothesis. The authors mention in the discussion specifically attapulgite and chlorite? Why not other clay materials? Why not soils? Apart from the logical fallacies, conclusions drawn from this part seem a bit out of the blue and their connection to previous parts appears arbitrary.
- Also, the wood consumption has not been properly discussed. The results from that part are missing in the result section and should be brought into context with Gautam and Henderson and Oberst et al. which studied in length the wood consumption under different moisture levels and in different situational context.
- I am missing a more comprehensive discussion of other potential factors influencing termite preference in the given situation with regards to other clay properties. The authors started discussing this a bit when they mention fertility of vegetation patches in the desert, but this was not satisfactorily elaborated on.
- Also “N. E. Carey, D. S. Calovi, P. Bardunias, J. S. Turner, R. Nagpal, J. Werfel, Differential construction response to humidity by related species of mound-building termites, Journal of Experimental Biology 222 (2019). doi:10.1242/jeb.212274. “ should be considered and discussed.
- It should be very clear in the introduction, what is known about the influence of different clay materials; clays are also categorised according to their particle size which has not at all been discussed. The authors need to thoroughly justify their choice of clay materials. Why did they not look at also silt materials? Silt materials are often reported to be preferred over clay materials, since the shrinkage of silt is less than that of clay. Could this be related to the different purpose of using soil, clay, silt etc; of the different context (building material versus sheeting material, construction for the inner nest versus material used as outer shell etc) – this needs to be discussed, needs a foundation in the methods and then but properly introduced as well as well.

Additional comments

I believe that some of the figures are not necessary, eg 2B or 3B seem redundant (not much value in seeing the extra layer of soil in C, or better, no value in seeing the setup without the soil). Some figures may be combined; overall the paper could be more focussed; maybe more in the form of a letter or short paper to initiate discussions about the influence of different soil types and their properties on termite foraging decisions considering different situational context.

---

## Round 0.2 · accepted · Accept

In my opinion, the manuscript has been improved and is ready to be published. The authors clearly highlighted the context of their lab experiments and discuss potential factors that may influence termite preferences in environmental conditions. The choice of materials has been now clearly justified, and details on soil/clay used in the study have been provided (e.g., water-holding capacities). Prior relevant literature has been considered and discussed.